# Comparison of high-intensity interval training versus moderate-intensity continuous training in pulmonary rehabilitation for interstitial lung disease: a randomised controlled pilot feasibility trial

Dimitra Nikoletou ![ORCID],[1,2] Irina Chis Ster ![ORCID],[3] Carmen Y Lech,[3] Iain S MacNaughton,[3] Felix Chua,[4,5] Raminder Aul,[6] Paul W Jones[3]

For numbered affiliations see end of article.

**Correspondence to**
Dr Dimitra Nikoletou;
dinikole@sgul.ac.uk

## ABSTRACT

**Objectives** This study aimed to investigate the feasibility and efficacy of high-intensity interval training (HIIT) compared with moderate-intensity continuous training (MICT) in pulmonary rehabilitation (PR) for people with interstitial lung disease (ILD).

**Design** Single-centre, randomised controlled feasibility, pilot trial.

**Setting** Patients were recruited from the chest clinic of a tertiary ILD centre and attended circuit-based PR in the hospital's gym, followed by a personalised 6-month community programme.

**Participants** 58 patients, stratified per ILD type, were randomised into two groups: 33 to HIIT (18 males:15 females) (mean age (SD): 70.2 (11.4) years) and 25 to the MICT exercise mode (14 males:11 females) (mean age (SD): 69.8 (10.8) years).

**Interventions** 8-week, twice weekly, circuit-based PR programme of exercise and education, followed by a personalised 6-month community exercise programme.

**Outcome measures** Feasibility outcomes included staff-to-patient ratio and dropout rates per group. Primary outcome was the 6 min walk distance (6MWD). Secondary outcomes included the sniff nasal pressure, mouth inspiratory and expiratory pressures, handgrip and quadriceps strength and health status. Random-effects models were used to evaluate average variation in outcomes through time across the two groups.

**Results** The 6MWD peaked earlier with HIIT compared with MICT (at 4 months vs 5 months) but values were lower at peak (mean (95% CI): 26.3 m (3.5 to 49.1) vs 51.6 m (29.2 to 73.9)) and declined faster at 6 months post-PR. Secondary outcomes showed similar faster but smaller improvements with HIIT over MICT and more consistent maintenance 6 months post-PR with MICT than HIIT.

**Conclusions** HIIT is feasible in circuit-based ILD PR programmes and provides quick improvements but requires closer supervision of training and resources than MICT and benefits may be less well sustained. This would make it a less attractive option for clinical PR programmes.

## STRENGTHS AND LIMITATIONS OF THIS STUDY

⇒ This randomised controlled feasibility, pilot study, showed that high-intensity interval training (HIIT) is feasible in circuit-based pulmonary rehabilitation (PR) programmes.

⇒ Our results identify the point at which exercise reinforcement may be needed after completion of a HIIT programme.

⇒ Although it was possible to recruit enough patients with interstitial lung disease from a single site, not enough patients with idiopathic pulmonary fibrosis could be recruited to evaluate the effect of HIIT on this particular group.

⇒ We did not use supplementary oxygen during PR for patients other than those already on long-term oxygen therapy.

⇒ This study adds valuable information for the design of larger, definitive studies, on the peak and subsequent deterioration of outcomes over time depending on exercise mode.

A definitive, multicentre randomised controlled trial is required to address the role of HIIT in ILD.
**Trial registration number** ISRCTN55846300.

## BACKGROUND

Interstitial lung disease (ILD) is a broad term used to describe a diverse group of pulmonary conditions that cause fibrotic changes in the lung interstitium with subsequent reduction in lung compliance and oxygen diffusion capacity.[1 2] ILD is categorised into different types according to clinical, radiological and pathophysiological changes and multidimensional indexes have been created to predict mortality risk.[3 4] The clinical presentation includes breathlessness on exertion, dry and persistent cough, fatigue, progressive exercise

limitation and respiratory failure at the very severe stages.[5] Pharmacological treatment for patients with progressive fibrotic ILD is limited to only a few medications aiming to slow down further development of fibrosis[6] and lung transplantation, the only life-extending option, carries risks and is not available to all patients.[7 8]

Pulmonary rehabilitation (PR) is a non-pharmacological strategy that improves physical function and quality of life[5 9] but in patients with ILD the optimal training programme, especially for longer-term benefits, remains unclear.[5 10] Recent evidence suggests that patients with some types of ILD, such as asbestosis and idiopathic pulmonary fibrosis (IPF), may receive greater benefit from PR than others and effects may be more lasting in patients with milder disease.[11] The traditional exercise mode in PR is moderate-intensity continuous training (MICT) which involves continuous exercise for 30–60 min at intensities ranging from 60% to 80% maximum heart rate.[12] However, other training modalities may offer greater benefits.

High-intensity interval training (HIIT) is a modality characterised by intervals of high-intensity exercise followed by intervals of low intensity or rest.[13] There is growing evidence that HIIT improves a broad range of cardiovascular risk factors, including insulin sensitivity and peak oxygen uptake ($VO_2peak$),[14] and its safety has been evaluated in patients with cardiovascular and metabolic disease.[14–17] Respiratory patients, who cannot maintain exercise due to breathlessness, have found HIIT acceptable[18–20] and although its overall superiority over MICT has been questioned,[21] it has been used safely in patients with ILD preparing for lung transplantation.[22] However, HIIT has yet to be evaluated in the context of an ILD-tailored PR programme.

The aim of this pilot study was therefore to evaluate: (1) the feasibility of using HIIT in an ILD-PR programme; (2) determine the short-term and medium-term effects of HIIT on exercise capacity, respiratory and peripheral muscle strength, breathlessness and health status; and (3) explore responses in different types of ILD.

Some of the results of this study have been reported previously in abstract form.[23]

## METHODS
### Study design
Single-blind, randomised-controlled feasibility, pilot trial.

### Study population
Symptomatic patients over 18 years old, of all types and severity, apart from sarcoidosis, and with a respiratory-physician diagnosis of ILD from a tertiary referral hospital, were invited to participate. This included patients with IPF (diagnostic criteria consistent with the International Consensus statement[1]), connective tissue disease-related ILD, non-specific interstitial pneumonia, usual interstitial pneumonia and chronic hypersensitivity pneumonitis. Patients were eligible to participate if they were ambulant

and had symptoms of dyspnoea on exertion (grades 1 to 3 in the modified Medical Research Council breathlessness (mMRC) scale).[24] Patients on long-term oxygen therapy (LTOT) were also included. Exclusion criteria were a history of syncope on exertion, patients having an acute exacerbation, patients with sarcoidosis and severe comorbidities (such as musculoskeletal, neurological or cardiovascular problems) that would interfere with their ability to exercise.

Prior to being included in this study, all participants had full lung function tests, performed by specialised technicians in a lung function laboratory and CT scans were part of their clinical diagnostic process and management. Participants were monitored throughout the study by respiratory physicians. The usual medical care was not affected by participation in this study.

All participants provided written informed consent. The trial protocol was preregistered with the ISRCTN registry (ISRCTN55846300) and approved by the South East Coast-Surrey Ethics committee (14/LO/0149).

### Patient and public involvement
People with ILD were involved in the original design of this research and determined acceptability of the proposed interventions and likelihood of adherence to this programme. During the study, all participants were involved in the conduct of the research by giving regular feedback in the education-discussion part of the PR programme and via interviews. On completion of this study, some participants formed a local ILD- Support Group and viewed the final results for dissemination.

### Randomisation and blinding
Following baseline assessments, participants were randomised into two groups: exercise using MICT (MICT group) or exercise using HIIT (HIIT group). Participants were stratified by ILD type to ensure a balanced distribution of types between groups. They were placed into three subgroups: (1) idiopathic group (eg, IPF); (2) autoimmune group (connective tissue disease-related interstitial lung disease (CTD-ILD)) and (3) chronic hypersensitivity pneumonitis/Extrinsic Allergic Alveolitis group (CHP/EAA). The investigator performing all assessments and the statistician were blinded throughout the study.

### Intervention
#### The ILD-PR programme
This was a circuit-based 8-week, twice weekly, outpatient programme, including only people with ILD and consisting of an hour of exercise and an hour of education. Exercise was set, supervised and progressed by two experienced physiotherapists, according to a standardised protocol for each group allocation. Supplementary oxygen was not used during the sessions. Participants already on LTOT continued to exercise with oxygen as recommended but were not given additional oxygen. Aerobic exercises included treadmill walking, brisk walking between two cones, cycle ergometer, trampette and step ups. Strength

training exercises, upper and lower limb, were conducted using 'Thera-bands' and functional movements such as sit-to-stand and ball raises.

A minimum of 30 min were dedicated to aerobic training. Aerobic training intensity at the MICT group was set at 60% maximum heart rate (HRmax) and progressed each week. In the HIIT group, intensity was set at 80% HRmax and patients were instructed to exercise in dynamic intervals followed by low-intensity exercise or rest. The aim in this group was to increase the amount of time on the 'high-intensity' phase each week.

In the first week, the average time on HIIT (at 80% HRmax) for each aerobic exercise was 2.5 min, therefore HIIT was 12.5 min of the total 30 min session, and this time increased each week. Progression of training was determined by reducing time in the low-intensity phase of HIIT and/or increasing time in the target HR. Each participant's individualised programme was recorded on exercise logs during the sessions.

Heart rate and oxygen saturation were monitored using pulse oximeters and perceived exertion and breathlessness were recorded at set intervals using the modified 10-point Borg Scale and the Rate of Perceived Exertion Scale, respectively.[25 26]

The education session, on the second hour of the programme, was common for all participants and included presentations and discussion on topics important to patients with ILD, such as topics about breathing and breathlessness (mechanics of breathing, helpful/unhelpful breathing patterns, the Active Cycle of Breathing technique, useful positions for recovering, pacing), topics about ILD (what happens with ILD, cough, associated conditions-rheumatoid arthritis, gastro-oesophageal reflux disease, Sjögren's syndrome, Bird fancier's lung, IPF) and topics about managing the condition (why healthier lifestyle is important?, keeping good posture, barriers to activity/exercise, dietary advice, smoking, incontinence, relaxation information, mood, anxiety and breathlessness, self-management and goal setting).

On the first day of the programme all participants were given an 'ILD-Pulmonary rehabilitation booklet' designed by our team, which contained information and pictures of all exercises, an exercise diary and summary of the education topics (see online supplemental file 1). All participants were encouraged to continue exercising at home on the days they did not attend the programme.

### The 6-month programme

Following completion of the PR programme, participants were given a personalised exercise programme, starting at the level achieved at the end of PR, and were asked to continue to exercise three times per week as well as do daily walking. The plan was recorded in their personal booklet. The 6-month programme included outdoor walking in HIIT or MICT mode, as well as stretching and strength training exercises. Some participants joined their local gym or organised exercise classes to ensure participation in group exercise at least once per week and followed a home programme the remaining time. Others opted for home-based exercise only. Regardless of the choice, during the 6-month period, participants were self-monitoring and had regular phone calls (once per month) and at least one home visit by the research physiotherapists on the third month. The physiotherapists guided participants on filling-in the exercise diaries and progressing training depending on mode of exercise. The ILD-PR booklet was used as a resource for the home exercises and as a reminder of the educational topics.

### Outcomes

All assessments were performed by the same investigator (DN), using identical instructions and methods, at three time points; at baseline, at 8 weeks (post-PR) and at 6 months (post-community programme). The investigator was blinded throughout the study.

Feasibility outcomes included the proportion of eligible patients who consented to the study, attrition in each group, staff-to-patient ratio and supervision requirements, reasons for discontinuing the programme and adverse events.

The primary efficacy outcome was functional exercise capacity and was measured using the 6 min walk distance (6MWD) test as recommended by the American Thoracic Society (ATS) guidelines[27] using standardised instructions.

Secondary outcomes were:
1. Respiratory muscle function—sniff nasal pressure (SNIP) and the maximal inspiratory and expiratory pressures (PImax and PEmax) were recorded using a handheld portable respiratory muscle testing device (MicroRPM, CareFusion, Basingstoke, UK). A minimum of 10 attempts were made for each test, on at least one session. PImax was recorded from functional residual capacity and PEmax from total lung capacity
2. Peripheral muscle function—handgrip strength, assessed in the dominant hand, with the elbow at 90° angle, using a hydraulic hand dynamometer (JAMAR Hydraulic Hand Dynamometer, Homecraft Rolyan Ltd, Nottinghamshire, UK). Quadriceps strength (QUADS-DOM) and hip flexion (HIP-DOM) were assessed in sitting position, with the knee at 90° angle and back upright and supported using a handheld dynamometer (Lafayette Manual Muscle Tester Model 01163, Lafayette Instruments, Lafayette, Indiana, USA).
3. Health status, using the St George's Respiratory Questionnaire for IPF (SGRQ-I)[28] and the Hospital Anxiety and Depression (HAD) Scale.[29] Spirometry data and anthropometric measurements (height, weight, %fat, waist, hip and neck circumference) were also collected.

Data were collected and managed using REDCap electronic data capture tools hosted at the NHS Trust.[30]

## Statistical analysis

This was a pilot study to test feasibility of the HIIT exercise mode and inform the design and power of a larger, next phase, randomised controlled trial. Intention-to-treat analyses are presented, that is, all participants who were randomised were considered in the analysis, not only those who completed the programme together with estimates' uncertainties as 95% CIs. This is not a hypothesis testing setting, hence the numbers followed instructions from the literature.[31] An indicative sample size calculation was based on previous research aiming to detect between-group changes of 38±43 m in the 6MWD following rehabilitation,[10 32 33] with an 80% power and a probability of type 1 error of 0.05.

Exploratory analyses used random effects (mixed) models to estimate average values in outcomes across the follow-up time for the four continuous longitudinal outcomes of interest, that is, 6MWD, SNIP, SGRQ-I and Quads strength which were relatively normally distributed on each occasion. Average estimates of these measurements were derived following fitting a series of a non-linear, quadratic mixed models over time allowing for a three-way interaction between time, intervention (HIIT vs MICT) and clinical group (ILD type) as well as between age, intervention and clinical group. For each of the four key exploratory outcomes of interest, a series of nested models of increasing complexity were developed and tested for their statistical improvement against the next less complex model. The complete data analysis (using missing data jargon) using mixed modelling on the four continuous longitudinal outcomes accounted for all participants and their complete observations (intention to treat analyses) and operated under 'missing at random' assumption for the attrition at the follow-up assumption which is not testable from the data in hand.[34] Although the missing data are less important in such pilot/feasibility settings,[31] we did investigate its patterns, particularly to assess potential differences between groups defined by intervention and /or clinical characteristics. Longitudinal binary outcomes were defined as 1 for a missing observation and 0 otherwise—at the baseline and at the follow-up and for each outcome—and have been explored using longitudinal models with random effects. The results of these exploratory analyses are presented in online supplemental file 2.

The average variations of outcomes through time, within and between groups, are presented as means (95% CI) and descriptive statistics as mean (SD). All analyses were conducted on STATA software (V.11).

## RESULTS

### Patient flow

Sixty participants consented to the study but two participants withdrew prior to commencement of the PR programme. A total of 58 participants started the PR programme. Figure 1 is the Consolidated Standards of Reporting Trials diagram showing participant flow.

Baseline characteristics are shown in table 1. There were no differences between groups at baseline, except for DLCO which was significantly lower in the HIIT group.

### Feasibility of HIIT compared with MICT

HIIT was well tolerated in a circuit-based clinical PR programme but required closer supervision to monitor participants' target heart rate and avoid desaturation below 85%. There were no adverse events from HIIT during or after the PR programme; although attrition in the HIIT group was higher (42% compared with 28% in the MICT group) this did not appear to be due to a specific factor that could be directly attributed to the type of programme (figure 1).

### Change in exercise capacity, respiratory and peripheral muscle strength and health status

The change in outcomes from baseline at all assessment points are shown in figure 2 and suggest a non-linear trend. This was then tested using linear and non-linear models and it was found that a quadratic model provided a significantly better fit than a linear model. The 6MWD increased over time from baseline in both groups, after which it tended to return towards starting levels (p<0.001 for a quadratic term of change over time), figure 3. The p value for the group–time interaction term was p=0.006 suggesting that the trajectories differed between groups. The HIIT group peaked quicker, around 4 months compared with around 5 months with MICT (figure 3). The models also suggested that the change in 6MWD values were smaller with HIIT (table 2, figure 2); mean change at peak was 51.6 m (95% CI: 29.2 to 73.9) in the MICT group compared with 26.3 m (95% CI: 3.5 to 49.1) with HIIT. The picture was the same using the estimates at 2 and 8 months (table 2, figure 2). Age was inversely associated with the average 6MWD values in both groups, but the time course was similar across ages (figure 3).

The quadratic term was significant for all secondary outcomes (p=0.0014 for SNIP, p=0.004 for QUADS-DOM and p=0.0014 for SGRQ-I) (figure 3).

SNIP values peaked earlier than 6MWD values in both groups by approximately 2 weeks (figure 3). QUADS-DOM values were maintained above baseline levels in both groups (figure 2) and showed a similar time course of change.

There was no evidence for a difference between treatment groups for the SGRQ-I values. The time to reaching the greatest improvement in SGRQ-I score was shorter in the HIIT group, but the changes in the HIIT group were of smaller magnitude at all time-points (online supplemental tables 1 and 2).

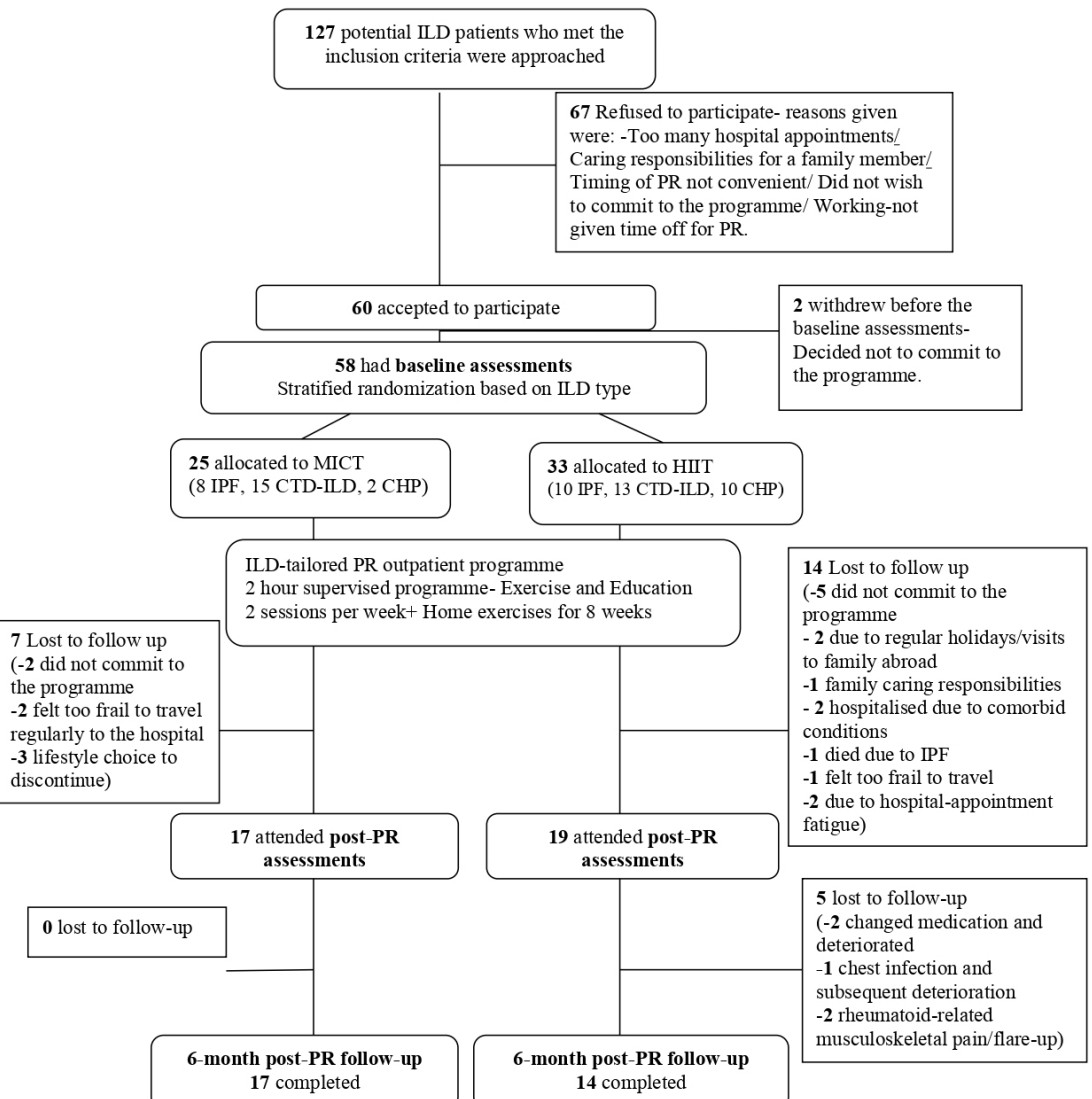

**Figure 1** Consolidated Standards of Reporting Trials diagram of participants' flow in the study. CHP, chronic hypersensitivity pneumonitis; CTD-ILD, connective tissue disease-related interstitial lung disease; HIIT, high-intensity interval training; ILD, interstitial lung disease; ILD-PR, pulmonary rehabilitation programme tailored for patients with interstitial lung disease; IPF, idiopathic pulmonary fibrosis; MICT, moderate-intensity continuous training; PR, pulmonary rehabilitation.

### Effects of HIIT on different aetiology ILD

Our models included three-way interactions between time, intervention (HIIT or MICT) and clinical group, to predict changes from baseline in the combined fibrosis (IPF and CHP group) and CTD-ILD groups. The patterns of change are consistent with the primary analysis, that is, while the HIIT group peaked faster than the MICT group, the level of improvement was less (online supplemental tables 1 and 2 and online supplemental figures 1–4).

### The effect of baseline DLCO on HIIT

Despite stratified randomisation, DLCO seemed to differ between intervention groups (table 1). We therefore performed additional, post-hoc analyses to examine its possible effect on outcomes. Based on these data, there was no evidence to suggest that DLCO further explained differences in the outcomes after accounting for time,

age at the recruitment, intervention and clinical groups and their interactions (see online supplemental file 3).

### DISCUSSION

To our knowledge, this is the first study to investigate the feasibility and potential efficacy of HIIT in the context of a circuit-based clinical PR programme for ILD and compare it with MICT. It demonstrated the feasibility of HIIT, in PR for patients with ILD and provided preliminary data on the exercise mode more likely to maintain benefits 6 months after programme completion. In addition, we identified the time of peak benefit and time course of subsequent deterioration.

HIIT proved to be feasible and well tolerated but required closer supervision and careful instructions for patients to continue this mode of training than MICT. The attrition rate was higher in the HIIT group compared

**Table 1** Baseline characteristics

| Variable | Total group (n=58) | Control group (MICT) (n=25) | Intervention group (HIIT) (n=33) | Between group P value |
|---|---|---|---|---|
| Age (years) | 70.0 (11.05) | 69.8 (10.8) | 70.5 (10.9) | 0.89 |
| Gender (male) (number (%)) | 32 (55.2) | 14 (56) | 18 (55) | 0.91 |
| ILD type (number (%)) | | | | |
| IPF | 18 (31.0) | 8 (32) | 10 (30.3) | 0.1 |
| CTD-ILD | 28 (48.3) | 15 (60) | 13 (39.4) | |
| CHP | 12 (20.7) | 2 (8) | 10 (30.3) | |
| ILD-GAP index (number (%)) | | | | |
| 0–1 | 24 (41.4) | 12 (48) | 12 (48) | 0.72* |
| 2–3 | 20 (34.5) | 9 (36) | 11 (33.3) | |
| 4–5 | 10 (17.2) | 4 (16) | 6 (18.2) | |
| >5 | 2 (3.4) | 0 (0) | 2 (6.1) | |
| BMI (kg/m$^2$) | 27.3 (4.6) | 27.5 (3.8) | 27.2 (5.1) | 0.84 |
| Waist/hip ratio | 0.9 (0.1) | 0.9 (0.1) | 0.9 (0.1) | 0.6 |
| Fat (%) | 29.9 (8.6) | 29.9 (9.3) | 30.1 (8.1) | 0.95 |
| FVC (%, predicted) | 79.1 (21.8) | 85.8 (24.0) | 73.8 (18.8) | 0.11 |
| DLCO (%, predicted) | 47.0 (14.7) | 53.9 (13.1) | 41.7 (13.9) | 0.002 |
| Comorbidities (number (%)) | | | | |
| Cardiac | 21 (36.2) | 8 (32) | 13 (39.4) | 0.56 |
| Hypertension | 23 (39.7) | 9 (36) | 14 (42.4) | 0.62 |
| Diabetes | 15 (25.9) | 7 (28) | 8 (24.2) | 0.75 |
| High cholesterol | 12 (20.7) | 7 (28) | 5 (15.2) | 0.23 |
| Prednisolone medication (number (%)) | | | | |
| <5 mg | 19 (32.8) | 9 (36) | 10 (30.3) | 0.34* |
| 5–10 mg | 19 (32.8) | 5 (20) | 14 (42.4) | |
| >10 mg | 3 (5.17) | 2 (8) | 1 (3) | |
| Smoking status (number (%)) | | | | |
| Current smoker | 13 (22.4) | 5 (20) | 8 (24.2) | 0.31* |
| Former smoker | 9 (15.5) | 6 (24) | 3 (9.1) | |
| Never smoked | 33 (56.9) | 13 (52) | 20 (60.6) | |
| Missing records | 3 | 1 | 2 | |
| Pack year history | 8.9 (15.9) | 10.7 (16.5) | 7.4 (15.6) | 0.5 |
| Median/IQR interval | 0 (0, 10) | 0 (0, 20) | 0 (0, 4.5) | |
| Missing | 7 | 3 | 5 | |
| 6MWD (m) | 371.7 (122.2) | 380.8 (139.8) | 364.6 (108.3) | 0.65 |
| 6MWD missing | 6 | 2 | 4 | |
| SpO$_2$ in room air (%) | 96 (1.5) | 96.5 (1.8) | 96 (1.5) | 0.3 |
| Missing | 6 | 2 | 4 | |
| No. on LTOT | 3 | 1 | 2 | n/a |
| SNIP (cmH$_2$O) | 94.8 (24.3) | 96.8 (25.9) | 93.1 (24.5) | 0.61 |
| Missing | 9 | 3 | 6 | |
| PImax (cmH$_2$O) | 91.5 (30.4) | 91.1 (34.2) | 91.8 (27.4) | |
| Missing | 8 | 2 | 6 | |

Continued

**Table 1** Continued

| Variable | Total group (n=58) | Control group (MICT) (n=25) | Intervention group (HIIT) (n=33) | Between group P value |
|---|---|---|---|---|
| PEmax (cmH$_2$O) | 106.6 (32.0) | 114.4 (40.3) | 100.7 (23.5) | 0.22 |
| Missing | 23 | 10 | 13 | |
| Handgrip (dominant hand) (kg) | 25.8 (10.2) | 27.2 (11.2) | 24.8 (9.4) | 0.42 |
| Handgrip missing | 4 | 2 | 2 | |
| Quads extension (dominant side) (kg) | 18.3 (5.5) | 18.7 (6.2) | 18 (5.0) | 0.65 |
| Missing | 2 | 0 | 2 | |
| Hip flexion (dominant side) (kg) | 15.6 (4.6) | 15.3 (5.1) | 15.9 (4.3) | 0.41 |
| Missing | 1 | 0 | 1 | |
| HAD-A | 6.3 (4.2) | 6.2 (4.6) | 6.3 (3.9) | 0.88 |
| Missing | 1 | 0 | 1 | |
| HAD-D | 5.5 (3.7) | 5.8 (3.2) | 5.2 (4.0) | 0.28 |
| Missing | 1 | 0 | 1 | |
| SGRQ-I total score | 46.6 (21.2) | 43.3 (20.3) | 49.3 (21.8) | 0.29 |
| Missing | 1 | 0 | 1 | |

Data are means (SD) unless otherwise stated. The p value tests the general null hypothesis of similarity between the randomised groups. The methods include t-tests (equal or unequal variances), Kruskal-Wallis and $\chi^2$ (Fisher's exact test) as appropriate according to the nature of the variables. All tests have been carried out on the complete data only and missing data were not considered as a separate category.
* P value was calculated considering the whole (balanced) distribution.
BMI, body mass index; CHP, chronic hypersensitivity pneumonitis; CTD-ILD, connective tissue disease-related interstitial lung disease; DLCO, Diffusion capacity of lung for carbon monoxide; FVC, forced vital capacity; HAD-A, Hospital Anxiety and Depression scale- Anxiety total score; HAD-D, Hospital Anxiety and Depression scale- Depression total score; ILD, interstitial lung disease; ILD-GAP index, Interstitial Lung Disease-Gender (G), Age (A) and lung physiology (P) variables (FVC and DLCO) point-scoring system.; IPF, idiopathic pulmonary fibrosis; LTOT, long-term oxygen therapy; 6MWD, 6 min walk distance; PEmax, maximal expiratory pressure; PImax, maximal inspiratory pressure; SGRQ-I, St George's Respiratory Questionnaire for IPF; SNIP, sniff nasal pressure.

with the MICT group and more contact was needed in the 6-month period to ensure that the HIIT mode of training met the protocol requirements.

The preliminary efficacy of HIIT over MICT was tested using random effects models. Their aim was to understand the average variation of the four key outcomes through time across groups to answer five main questions: (1) How do these measurements change over time? (2) Does the benefit differ between groups? (3) How sustainable is the benefit? (4) Does the attrition rate differ between groups? and (5) What is the influence of age? The study was primarily designed to compare the two treatment groups, but an exploratory analysis was performed to look for differences between clinical groups, although this resulted in small numbers in the subgroups.

Contrary to our initial hypothesis, our data showed no evidence of greater benefit with HIIT compared with that achieved with MICT in exercise capacity, respiratory and peripheral muscle function or health status in the medium term. The study was designed as a pilot to test the potential for a trial of HIIT in a larger, definitive study, but we are forced to conclude that a study of the same design may not be worthwhile.

At 2 months, both groups improved >34 m in the 6MWD, which is the reported minimal clinical important

difference in patients with IPF,[10 35 36] adding to existing evidence on the benefits of PR in ILD.[10 11 32 37–41] However, although the 6MWD improved earlier in the HIIT group (peaking at 4 months instead of 5 months) differences from baseline were consistently lower and not as well maintained at 8 months compared with the MICT group. A similar pattern emerged when the effect of age was added to the model. Age is an important factor in indices associated with mortality risk in ILD[3 4] and our results showed lower 6MWD with greater age. Despite this age effect, PR had a positive effect in all age groups regardless of baseline levels, contrary to previous studies reporting greater improvement in patients with lower baseline 6MWD.[11 36 40 42] However, HIIT did not offer additional or preferential benefits to any specific age group and tended to decline below baseline levels at 8 months in the older group. In contrast, the benefit seen in the older MICT group was maintained above baseline levels.

In addition to functional exercise capacity, patients improved quadriceps muscle strength post-PR but similarly, there was no difference between treatment groups. Age-adjusted results for quadriceps force were less variable than for 6MWD and benefits were well maintained above baseline in both groups. The modelling suggested that quadriceps force peaked a little later than other outcomes

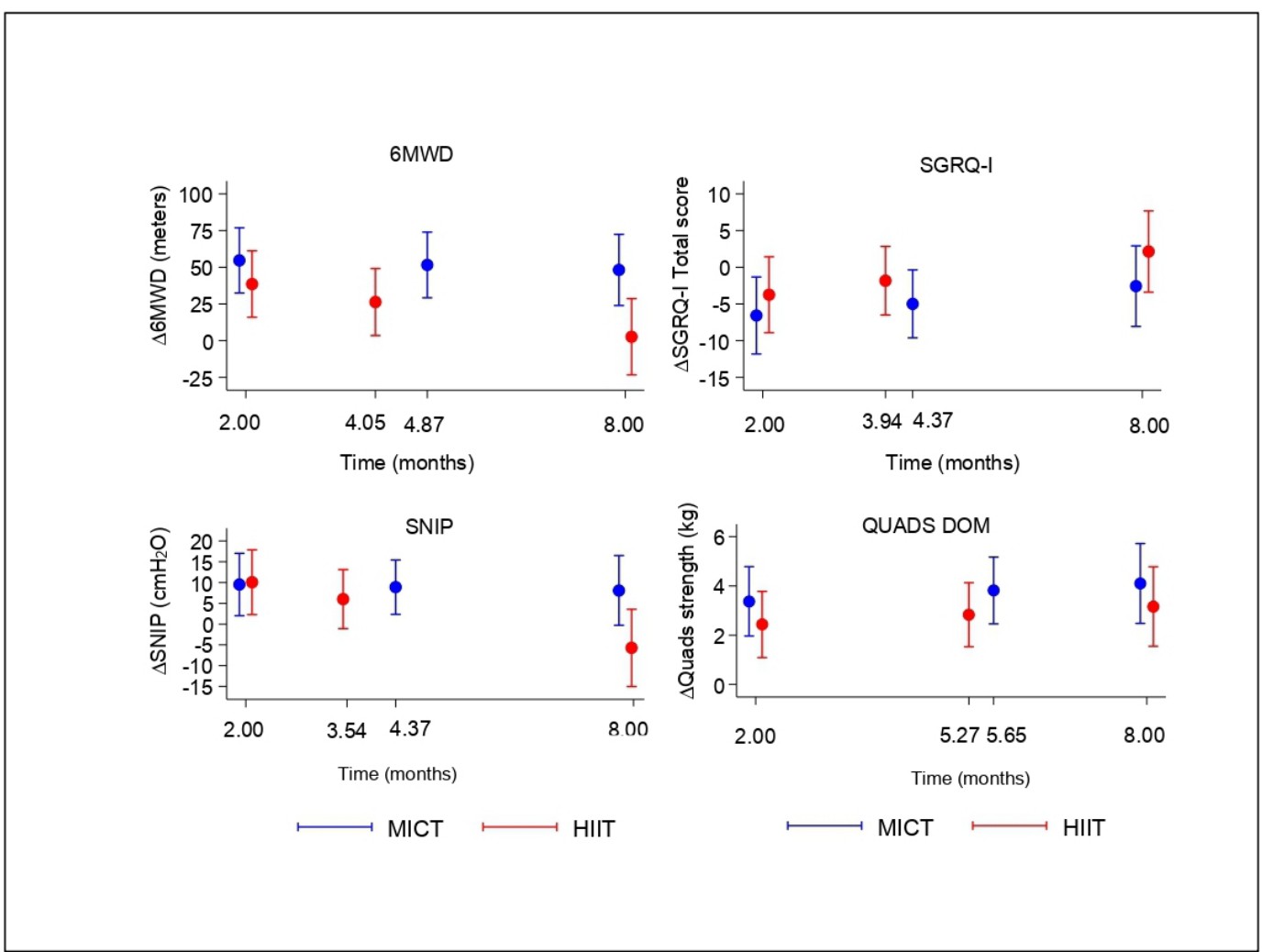

**Figure 2** Change from baseline in four key outcomes by intervention group over time. 6MWD, 6 min walk distance; HIIT, high-intensity interval training; MICT, moderate-intensity continuous training; SNIP, Sniff nasal pressure.

at 5.5 months in the MICT group and at 5 months at the HIIT group. Our results support the inclusion of strength training in ILD-PR programmes and suggest that it may explain why some studies showed greater improvement and maintenance in functional capacity[32] while others, that focused on endurance training, did not.[11] The time course of peak improvement and subsequent decline in quadriceps strength did not follow the pattern seen with 6MWD, in either group. Quadriceps strength has been shown to deteriorate in line with ageing and severity in other chronic respiratory disorders[43] and has been identified as an important predictor of mortality in Chronic Obstructive Pulmonary Disease (COPD).[44] Skeletal muscle atrophy has also been identified in advanced ILD.[45] Our ILD group, which included people in advanced stage (as defined by the ILD-GAP score >5, table 1) showed improvement in mean quadriceps force post-PR in both groups and across all age groups. We found no evidence that older individuals benefited more from PR.

Respiratory muscle strength was preserved at baseline (mean SNIP, MIP and MEP >70 cmH$_2$O in both groups) as previously described[46 47] and improved post-PR in both

groups even though there was no inclusion of specific inspiratory muscle training in our programme. In addition, the SNIP followed a similar pattern over time to 6MWD and peaked approximately 2 weeks earlier than the 6MWD in both the HIIT and MICT group. Although there was no additional benefit with HIIT, our results suggest that SNIP may be more useful in monitoring changes following PR than quadriceps force in patients with ILD. In support of this conclusion, Mendoza and colleagues[46] showed that inspiratory muscle strength correlated better with 6MWD than quadriceps twitch force in patients with fibrotic idiopathic interstitial pneumonia. Although our ILD group was more mixed and included patients with CTD-ILD, our results confirm that respiratory muscle strength appears preserved in ILD and that inspiratory muscle strength correlates well with the 6MWD. Our study is the first to show that inspiratory muscle strength changes in tandem with the 6MWD following PR, regardless of the chosen mode of training.

Health status, as defined by the SGRQ-I and HAD Scale improved post-PR and was maintained at 8 months, but

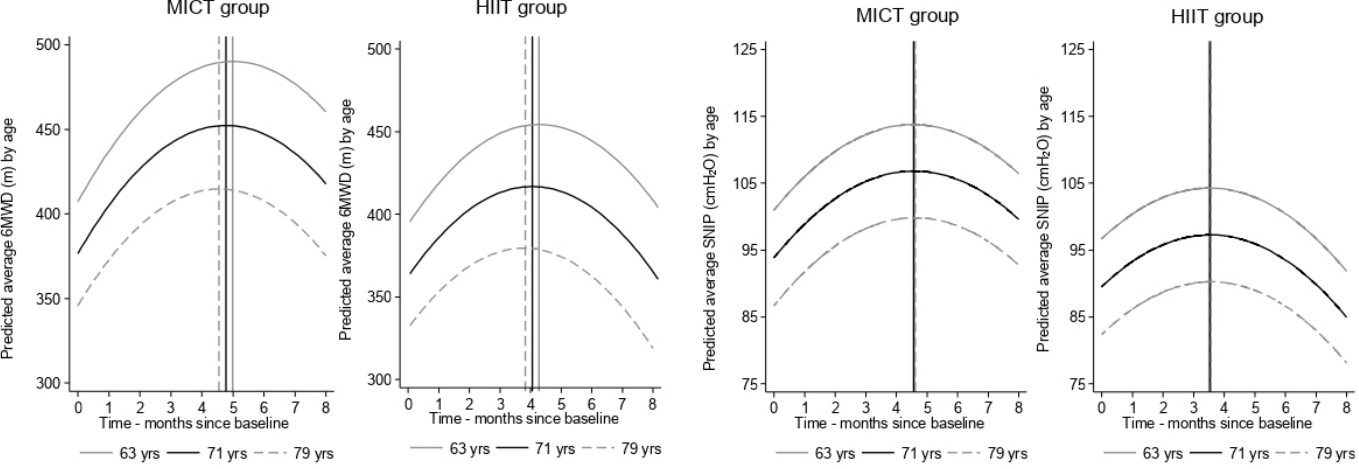

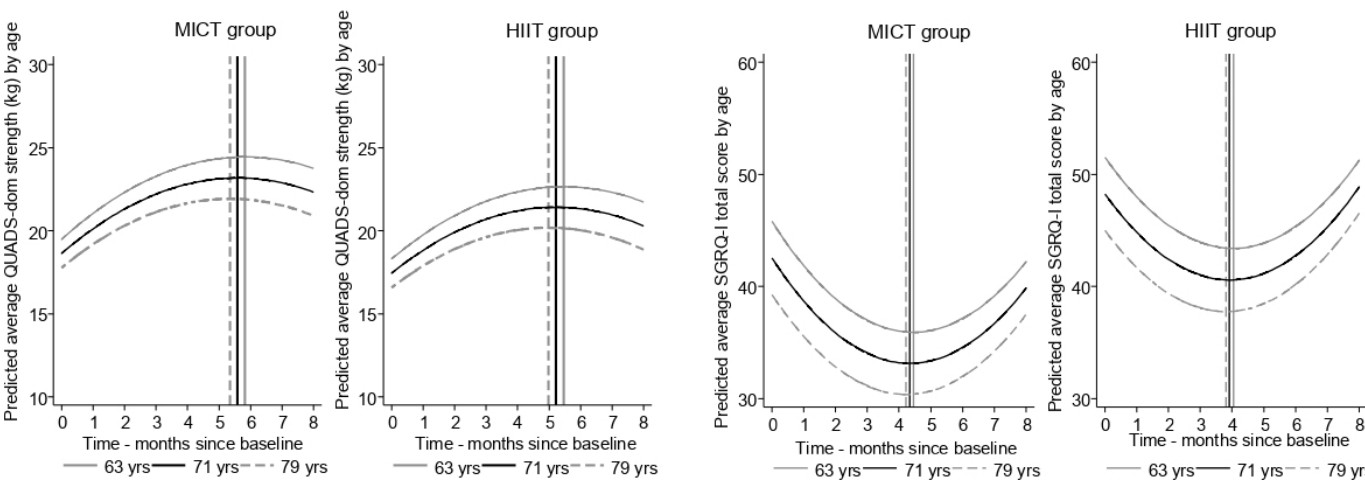

**Figure 3** Age-related variations in the four key outcomes (6MWD, SNIP, QUADS-DOM (quadriceps strength) and SGRQ-I total score) per training group using random effects models. The vertical line indicates the inflection point (ie, highest or lowest values) in each age group. The ages are the mean values in each tertile. 6MWD, 6 min walk distance; HIIT, high-intensity interval training; MICT, moderate-intensity continuous training; SGRQ-I, St George's Respiratory Questionnaire for IPF; SNIP, sniff nasal pressure.

again, there was no evidence of superiority of HIIT over the MICT group (online supplemental table 1).

### Effects of HIIT on different aetiology ILD

Random-effect models were also created based on ILD type (clinical subgroups). We combined the IPF and CHP subgroups into the combined 'Fibrosis' clinical group. This was due to smaller numbers of IPF participants at the end of the study which did not allow for independent evaluation of this subgroup.

The effects of HIIT on ILD of different aetiology were consistent with our primary analysis, that is, that HIIT peaked faster and the level of improvement was less. This result seemed to be largely attributable to the fibrosis group, which peaked fastest but showed the lowest improvement from baseline compared with the CTD-ILD group. Similar findings were observed for the other outcomes, again showing lack of preferential

benefit from HIIT in any subgroup. While the CTD-ILD subgroup improved in most variables, the HIIT mode was not superior to MICT.

### LIMITATIONS

The study has limitations. First, although our retrospective analysis showed no evidence that baseline DLCO had an effect on outcomes following our training programme (online supplemental file 3), the effect may have been concealed by the size of this pilot study. Imbalance in randomisation can happen by chance in clinical trial settings, particularly in pilot/feasibility studies.[48] However, DLCO did not affect PR benefits in a previous study,[49] but future definitive studies may need to consider controlling this factor at randomisation.

**Table 2** Average predictions for main outcomes values at the main time points during the trial and for changes from baseline (actual and adjusted for age)

| Outcome | Time | Group data Mean (95% CI) | | Change from baseline Mean (95% CI) | | Change from baseline adjusted for age Mean (95% CI) | |
|---|---|---|---|---|---|---|---|
| | | Control (MICT) | Intervention (HIIT) | Control (MICT) | Intervention (HIIT) | Control (MICT) | Intervention (HIIT) |
| 6MWD (m) | Baseline | 382 (336, 429) | 372 (331, 412) | | | | |
| | 2 months | 433 (386, 480) | 411 (370, 453) | 54.7 (32., 76.9) | 38.6 (15.95, 61.17) | 53.2 (33.66, 72.70) | 36.3 (16.42, 56.06) |
| | 8 months | 428 (378, 476) | 374 (331, 418) | 48.2 (24.0, 72.4) | 2.64 (−23.4, 28.6) | 46.6 (26.5, 66.7) | 0.06 (−21.95, 22.08) |
| | Peak value | 460 (411, 508) | 425 (382, 468) | 51.6 (29.2, 73.9) | 26.29 (3.5, 49.1) | 50.02 (31.2, 68.8) | 23.88 (4.5, 43.3) |
| | Peak time-months | 4.87 (4.43, 5.30) | 4.05 (3.62, 4.48) | | | | |
| SNIP ($cmH_2O$) | Baseline | 95 (85, 105) | 92 (82, 101) | | | | |
| | 2 months | 104 (94, 115) | 98 (88, 108) | 9.53 (2.02, 17.04) | 10.07 (2.28, 17.85) | 9.21 (1.98, 16.4) | 9.40 (1.88, 16.92) |
| | 8 months | 101 (90, 112) | 86.9 (76, 98) | 8.08 (−0.32, 16.48) | −5.73 (−15.03, 3.56) | 7.38 −1.05774 15.81 | ▲ 6.35 ▲ 15.64376 2.95 |
| | Peak value | 108 (97, 120) | 99 (89, 110) | 8.89 (2.33, 15.44) | 6.01 (−1.10, 13.12) | 8.39 (2.00, 14.78) | 5.36 (−1.53, 12.25) |
| | Peak time-months | 4.57 (3.72, 5.43) | 3.54 (2.57, 4.51) | | | | |
| QUADS DOM (kg) | Baseline | 18.8 (16.7, 20.9) | 17.6 (15.7, 19.5) | | | | |
| | 2 months | 21.6 (19.4, 23.8) | 20.1 (18.1, 22.1) | 3.37 (1.97, 4.78) | 2.44 (1.09, 3.78) | 3.33 (1.91, 4.74) | 2.39 (1.03, 3.74) |
| | 8 months | 22.7 (20.2, 25.6) | 20.6 (18.2, 23.0) | 4.10 (2.48, 5.72) | 3.16 (1.55, 4.78) | 4.06 (2.46, 5.65) | 3.12 (1.52, 4.71) |
| | Peak value | 23.5 (21.0, 26.0) | 21.7 (19.3, 24.1) | 3.82 (2.46, 5.17) | 2.83 (1.53, 4.13) | 3.77 (2.43, 5.11) | 2.78 (1.49, 4.07) |
| | Peak time-months | 5.65 (4.39, 6.91) | 5.27 (4.19, 6.36) | | | | |

**Table 2** Continued

| Outcome | Time | Group data Mean (95% CI) | | Change from baseline Mean (95% CI) | | Change from baseline adjusted for age Mean (95% CI) | |
|---|---|---|---|---|---|---|---|
| | | Control (MICT) | Intervention (HIIT) | Control (MICT) | Intervention (HIIT) | Control (MICT) | Intervention (HIIT) |
| SGRQ-I (total score) | Baseline | 43.2 (35.4, 51.1) | 48.8 (41.9, 55.7) | | | | |
| | 2 months | 36.6 (28.5, 44.7) | 43.0 (35.8, 50.2) | −6.57 (−11.81, to 1.33) | −3.75 (−8.92, 1.42) | −6.39 (−11.62, to 1.17) | −3.54 (−8.71, 1.63) |
| | 8 months | 40.3 (31.8, 48.8) | 49.2 (41.3, 57.1) | −2.57 (−8.06, 2.92) | 2.15 (−3.39, 7.68) | −2.40 (−7.89, 3.09) | 2.37 (−3.18, 7.93) |
| | Min value | 33.9 (25.1, 42.6) | 41.1 (33.1, 49.1) | −4.99 (−9.62 to 0.36) | −1.84 (−6.51, 2.83) | −4.82 (−9.44 to 0.19) | −1.63 (−6.31, 3.05) |
| | Min time-months | 4.37 (3.68, 5.07) | 3.94 (3.24, 4.65) | | | | |

HIIT, high-intensity interval training; MICT, moderate-intensity continuous training; 6MWD, 6 min walk distance; QUADS-DOM, quadriceps strength; SGRQ-I, St George's Respiratory Questionnaire for IPF; SNIP, sniff nasal pressure.

Second, we did not use supplementary oxygen during PR for participants other than for those already on LTOT. Although promising benefits have been reported with acute oxygen supplementation during exercise in patients with IPF,[50] there is still insufficient evidence about medium-term to long-term benefits in patients with ILD.[33] For our study, we felt it was important to train participants to recognise the time it took before desaturation occurred, so that they were able to continue that pattern of training and monitor themselves when unsupervised in community. Participants already on LTOT continued to exercise with oxygen. However, none of the three participants on LTOT finished PR, regardless of group allocation, supporting previous reports that PR may not be appropriate for them.[51] Future studies need to explore different models of training for these cases, such as home-based programmes.

Finally, the number of IPF participants who remained in the study was small and did not allow us to investigate the effect of HIIT over time in this particular group.

## IMPLICATIONS FOR FUTURE DEFINITIVE STUDIES

This pilot, enabled us to explore the effect of mode of training on an ILD-PR programme of 8 weeks duration followed by community-based training for 6 months. Our programme duration reflects current UK-based clinical programmes and showed improvements that were maintained at 8 months and were similar to a recent study of a much longer PR programme duration.[32] Although we conclude that there is no evidence to support the use of HIIT in a future definitive study of similar design to this pilot, our results suggest that HIIT may have a role when quicker improvements are required, such as in preoperative rehabilitation for ILD and that HIIT is feasible to be investigated further in a larger, definitive study of different design. Our results also provide valuable information on the time point where 6MWD and other variables start to deteriorate post-PR and therefore, an estimate of time when additional PR support may be needed, thus optimising PR provision for patients with ILD .

## CONCLUSIONS

In the current study, HIIT was found to be feasible and improved exercise capacity in an ILD-PR programme, but this exercise modality showed no evidence of superiority over MICT. Indeed, our findings suggest that it may be less effective and may require closer monitoring to avoid exercise-induced desaturation so, until further evidence becomes available, MICT appears to be more appropriate in the general clinical setting.

**Author affiliations**
[1]Centre for Allied Health, Institute of Medical and Biomedical Education, St George's University of London, London, UK
[2]Faculty of Health, Science, Social Care and Education, Kingston University, Kingston-Upon-Thames, UK

[3]Infection and Immunity Research Institute, St George's University of London, London, UK

[4]Interstitial Lung Disease Unit, Royal Brompton and Harefield NHS Foundation Trust, London, UK

[5]National Heart and Lung Institute, Imperial College London, London, UK

[6]Respiratory Medicine, St George's University Hospitals NHS Foundation Trust, London, UK

**Acknowledgements** The authors would like to thank all patients with ILD and particularly the St George's Interstitial Lung Disease Support Group for their participation and input in this study.

**Contributors** Concept and design of study: DN, PWJ, FC, RA, ICS. Acquisition of data: DN, CYL, ISM. Statistical analytical strategy: ICS. Analysis of data: DN, ICS. Drafting of manuscript: DN, ICS, PWJ. Revision of manuscript critically for important intellectual content: all authors. Approval of final manuscript: all authors. DN is responsible for the overall content as the guarantor.

**Funding** The study was funded by the National Institute for Health Research (NIHR) Research for Patient Benefit (RfPB) Programme (Grant Reference Number PB-PG-1112-29067). The views expressed are those of the authors and not necessarily those of the NHS, the NIHR or the Department of Health.

**Competing interests** DN, ISM and CYL declare no financial interests. ICS has had grants from NIHR, Astra-Zeneca and BHF and is a statistical expert for the Health Research Authority. FC has received speaker fees and financial sponsorship for conducting meetings by Boehringer-Ingelheim and Roche. RA has received lecture fees from Pfizer and Novartis. PWJ owns GlaxoSmithKline shares.

**Patient and public involvement** Patients and/or the public were involved in the design, or conduct, or reporting or dissemination plans of this research. Refer to the Methods section for further details.

**Patient consent for publication** Not applicable.

**Ethics approval** The ethics that approved the study was South East Coast-Surrey NRES committee. The reference number was 14/LO/0149. Participants gave informed consent to participate in the study before taking part.

**Provenance and peer review** Not commissioned; externally peer reviewed.

**Data availability statement** Data are available on reasonable request. All relevant data are presented in this paper; and more information can be provided on reasonable request from the corresponding author

**ORCID iDs**
Dimitra Nikoletou http://orcid.org/0000-0001-8229-3190
Irina Chis Ster http://orcid.org/0000-0003-2637-1259

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
