## [Reviewer comments · BMJ Open]

ARTICLE DETAILS

TITLE (PROVISIONAL)	Comparison of high intensity interval training versus moderate intensity continuous training in pulmonary rehabilitation for interstitial lung disease; a randomised controlled pilot feasibility trial.
AUTHORS	Nikoleitou, Dimitra; Chis Ster, Irina; Lech, Carmen; MacNaughton, Iain; Chua, Felix; Aul, Raminder; Jones, Paul

VERSION 1 – REVIEW

REVIEWER	Moy, Marilyn L. Brigham
REVIEW RETURNED	30-Aug-2022

GENERAL COMMENTS	General Comments This is an interesting single site RCT pilot study of high intensity interval training compared to moderate intensity in pulmonary rehab for patients with ILD. The statistical analysis is difficult to understand and interpret given that the models were quadratic equations. The lack of adjustment in the analysis for the imbalance of randomization is a weakness. Models should adjust for diffusion capacity given that it was significantly different between groups. Also, why are the group sizes so different when there was 1:1 randomization? Lack of information about engagement with the intervention is a major weakness. We know how many patients dropped out at each time point and did not have assessments, but what is known about how they engaged with the intervention? Also, significantly more participants dropped out of the high intensity intervention, so it is unclear whether the conclusion that it is feasible is substantiated. More details of the intervention are needed particularly the duration of intervals. The introduction and discussion should provide succinct context of the existing literature of the efficacy of moderate intensity training in PR for ILD and what gap/unanswered question this study examines. Specific Comments Abstract Participant sex and age by group are missing. Methods Was a test 6MWD performed, as recommended by ATS guidelines? Were PFTS (i.e.lung volumes) or CT scans used to assess the
---

	degree of restriction and extent of disease? Provide details of the education session topics. Results Table 2 would be easier to understand if presented as a figure.
--	--

REVIEWER	k, vaishali Manipal College of Health Professions, Physiotherapy
REVIEW RETURNED	14-Oct-2022

GENERAL COMMENTS	This is a very well written manuscript. Although I have a few concerns. 1) In the Methods section,  - it would be good to provide operational definition of the study population. - in the eligibility criteria, the authors have mentioned "Patients were eligible to participate if they were ambulant and had symptoms of dyspnoea on exertion." what grade of dyspnoea was acceptable for inclusion in the study? - Were participants having acute exacerbations also included? if yes, what was the criteria for inclusion. - The authors mentioned participants with LTOT in the results section. I think it would be good if it was mentioned initially in the eligibility criteria section for a clear perspective. - Did the authors decide regarding not including participants using supplementary oxygen during the PR program? I think inclusion of the same in the eligibility section would be necessary. 2) The authors have mention "On the first day of the programme all participants were given an 'ILD-Pulmonary rehabilitation booklet' designed by our team, which contained information and pictures of all exercises, an exercise diary and summary of the education topics." . Was the educational booklet adapted from anywhere? if yes, it can be referenced accordingly. If not, how was the booklet developed, a short description regarding development and validation process should be mentioned in the methods section. 3) How was the sample size determined? Detailed explanation regarding how the total number of ILD patients was derived needs to be mentioned. Were the drop-out rates considered? 4) How was the missing data managed? 5) The authors mention the study period for as "n 8-week, twice weekly outpatient programme" followed by a 6 month home exercise program. What was the schedule/ plan for the home exercise program? Did the patients perform the exercises all by themselves referring to the educational booklet or were monitored? If yes, how?
--

REVIEWER	Wang, Zhijie Guangzhou University of Chinese Medicine, Medical College of Acupuncture and Rehabilitation
REVIEW RETURNED	03-Feb-2023

GENERAL COMMENTS	1. Please add the information on how to do HIIT and MICT in the interventions part of abstract. 2. The authors mentioned "Pulmonary rehabilitation (PR) is a well-established rehabilitation option for various respiratory conditions [5, 6] but in ILD patients the optimal training programme, especially for longer term benefits, remains unclear. ". are there other therapies for ILD? What are the effects?
---

	3. Lack of the information on why the authors did this study. 4. The authors just introduce HIIT in the background section, but what is MICT? 5. The aim of this pilot study seems not to match the title. 6. How did the authors do the randomization? 7. Please give a reference for MICT was 60% maximum heart rate and HIIT was 80%. Or please give a clear definition for MICT and HIIT. 8. The authors mentioned “The average time on HIIT was 2.5 min in the first week and recovery time varied depending on type and severity of ILD”, so how long is that and how to recommend for more patients in the clinic? 9. Who and where did the patients monitor the pulse oximeters and perceived exertion and breathlessness? 10. How to calculate the sample size? 11. Lack of the information on severity of ILD of the included patients. 12. The authors mentioned all types and severity of ILD would be included in this study, so I just want to know how the authors did the risk monitoring and what they did for the potential danger? 13. How about to take medicines for the comorbidities? 14. Did the included patients take medicines for ILD? 15. The reference would be up-to-date.
--	--

VERSION 1 – AUTHOR RESPONSE

Comments on:

(Questions and suggestions from Reviewer 1)

- 1. The statistical analysis is difficult to understand and interpret given that the models were quadratic equations. The lack of adjustment in the analysis for the imbalance of randomization is a weakness.**

Thank you for this comment – please find our views below.

We appreciate the issues raised using a quadratic model. Nevertheless, they are increasingly adopted in these stances due to at least two reasons. The measurements are not taken at equally spaced times and depend on one another and the attrition process. Simple tests (paired or unpaired) would only have to consider participants with complete observations to provide valid results, hence missing important information. The analyses based on mixed modelling include all available observations and operate under missing at random assumption (doi: <https://doi.org/10.1136/bmj.b2393>) which is not testable from the data at hand. But the likelihood-based estimation takes into account all available observations regardless of their number are say 30 at the baseline, 20 on the second occasion and 15 or 25 on the third occasion. As mentioned before, it is incorrect to ignore a quadratic term when it explains variability in the data and, more importantly, has important implications in data interpretation.

Feasibility studies and external pilot studies are increasingly common precursors to a main, or definitive, randomized controlled trial (<https://doi.org/10.1186/1471-2288-10-1>, <https://doi.org/10.1111/j.2002.384.doc.x>). As we have mentioned many times in the manuscript, these studies may address process measures, such as the number of eligible patients in a centre, the consent rate, rates of treatment fidelity and compliance, and the methods of randomization, blinding

and outcome measurement - but they are not hypothesis testing studies, and we are not claiming this at any point.

The reviewer is right pointing to the imbalance of the randomization. Nevertheless, this can happen by chance even in clinical trials settings, particularly when small studies are featured ([10.1136/bmj.319.7203.185](https://doi.org/10.1136/bmj.319.7203.185)).

Adequate tests have been applied for the baseline variables (secondary exploratory outcomes not included though – such as *diffusion capacity* to which the reviewer particularly points to- as they were part of the pilot exploratory analysis. As such, t-tests (equal or unequal variances), Kruskal-Wallis and chi-squared (or Fisher exact) did not reveal any differences in the demographics, namely age, sex, ILD-GAP index distribution across the two groups did not reveal statistical imbalance. Table 1 in the manuscript (also displayed below for the reviewers' perusal) shows the values per group and between group differences. Any potential significant difference is also down to chance in case of randomized experiments.

In conclusion, there is no evidence of statistical imbalance between the two groups regarding the baseline measurements.

Variable	Total Group p (n= 58)	Control group p (MICT) (n=25)	Intervention group (HIT) (n=33)	Between group P value
Age (years)	70.0 (11.05)	69.8 (10.8)	70.5 (10.9)	0.89
Gender (male) (number (%))	32 (55.2)	14 (56)	18 (55)	0.91
ILD Type (number(%))				0.10
 • IPF • CTD-ILD • CHP 	18 (31.0) 28 (48.3) 12 (20.7)	8 (32) 15 (60) 2 (8)	10 (30.3) 13 (39.4) 10 (30.3)	
ILD-GAP index (number (%))				0.72
 • 0-1 • 2-3 • 4-5 • >5 	24 (41.4) 20 (34.5) 10 (17.2) 2 (3.4)	12 (48) 9 (36) 4 (16) 0 (0)	12 (48) 11 (33.3) 6 (18.2) 2 (6.1)	
BMI (kg/m ²)	27.3 (4.6)	27.5 (3.8)	27.2 (5.1)	0.84
Waist/Hip ratio	0.9 (0.1)	0.9 (0.1)	0.9 (0.1)	0.60
fat (%)	29.9 (8.6)	29.9 (9.3)	30.1 (8.1)	0.95
FVC (%predicted)	79.1 (21.8)	85.8 (24.0)	73.8 (18.8)	0.11
DLCO (% predicted)	47.0 (14.7)	53.9 (13.1)	41.7 (13.9)	0.002
Comorbidities (number (%))				0.56
 • Cardiac • Hypertension • Diabetes • High Cholesterol 	21 (36.2) 23 (39.7) 15 (25.9) 12 (20.7)	8 (32) 9 (36) 7 (28) 7 (28)	13 (39.4) 14 (42.4) 8 (24.2) 5 (15.2)	0.62 0.75 0.23
Prednisolone medication (number(%))				
 • < 5mg 	19 (32.8)	9 (36)	10 (30.3)	0.34
 • 5-10 mg 	19 (32.8)	5 (20)	14 (42.4)	

• >10 mg	3 (5.17)	2 (8)	1 (3)	
Smoking status (number (%))	13 (22.4)	5 (20)	8 (24.2)	0.31
• Current smoker	9 (15.5)	6 (24)	3 (9.1)	*Missing data are NOT interpreted as a category
• Former smoker	33 (56.9)	13 (52)	20 (60.6)	
• Never smoked	3	1	2	
• Missing records				
Pack year history	8.9 (15.9)	10.7 (16.5)	7.4 (15.6)	0.50
Median/IQR interval	0(0, 10)	0(0, 20)	0(0, 4.5)	
Missing	7	3	5	
6MWD (m)	371.7 (122.2)	380.8 (139.8)	364.6 (108.3)	0.65
Missing	6	2	4	
SpO2 in room air (%)	96 (1.5)	96.5 (1.8)	96 (1.5)	0.3
Missing	6	2	4	
No on LTOT (number)	3	1	2	NOT SURE WHAT THAT IS
SNIP (cmH ₂ O)	94.8 (24.3)	96.8 (25.9)	93.1 (24.5)	0.61
Missing	9	3	6	
Plmax (cmH ₂ O)	91.5 (30.4)	91.1 (34.2)	91.8 (27.4)	
Missing				
PEmax (cmH ₂ O)	106.6 (32.0)	114.4 (40.3)	100.7 (23.5)	0.22
Missing	23	10	13	
Handgrip (dominant hand) (kg)	25.8 (10.2)	27.2 (11.2)	24.8 (9.4)	0.42
Missing				
Quads extension (dominant side) (kg)	18.3 (5.5)	18.7 (6.2)	18 (5.0)	0.65
Missing	2	0	2	
Hip flexion (dominant side) (kg)	15.6 (4.6)	15.3 (5.1)	15.9 (4.3)	0.41
	1	0	1	
HAD-A	6.3 (4.2)	6.2 (4.6)	6.3 (3.9)	0.88
	1	0	1	
HAD-D	5.5 (3.7)	5.8 (3.2)	5.2 (4.0)	0.28
	1	0	1	
SGRQ-I Total score	46.6 (21.2)	43.3 (20.3)	49.3 (21.8)	0.29
	1	0	1	

2. Models should adjust for diffusion capacity given that it was significantly different between groups.

We would like to thank this reviewer for raising this very valid point.

We have now investigated DLCO in a number of different ways.

1. Baseline assessment, i.e. DLCO by intervention and clinical groups simultaneously.

Based on these data, the evidence suggests that DLCO differs by intervention group (adjusted p value=0.003)– finding initially deems as random randomization imbalance. Nevertheless, there is not enough evidence to suggest that DCLO levels differ by clinical group, i.e. the DCLO difference between the clinical groups seems to remain similar with the intervention group (p=0.069).

2. Relationship between the 4 outcomes and DLCO at the baseline. The results are presented in Table 1 below. There was no evidence for an interaction of DLCO with either intervention or clinical group regarding these outcomes. Only coefficients corresponding to DLCO are shown for a succinct message, but they are adjusted for intervention groups and clinical groups.

Table 1: Relationship between the 4 key outcomes and baseline DLCO, adjusted for intervention and clinical groups.

	Estimate	SE	T value	p-value	95%CI low	95%CI high	No Obs	Association
MWT	.9787011	1.178773	0.83	0.410	-1.39138	3.348783	50	UNADJUSTED
SNIP	.1441871	.2431136	0.59	0.556	-.3454688	.633843	47	
QUADS DOM	-.0956757	.0499201	-1.92	0.061	-.1958477	.0044964	54	
SGRQI	-.0614169	.1979514	-0.31	0.758	-.4586354	.3358016	54	
MWT	8799743	1.31952	0.67	0.508	-1.774558	3.534506	50	ADJUSTED FOR
SNIP	.0873187	.2717802	0.32	0.750	-.4604183	.6350556	47	INTERVENTION
QUADS DOM	-.123836	.0548133	-2.26	0.028	-.2338782	-.0137938	54	GROUP
SGRQI	.0017764	.2204537	0.01	0.994	-.4408028	.4443556	54	
MWT	1.047737	1.366581	0.77	0.447	-1.703048	3.798522	50	ADJUSTED FOR
SNIP	.1784858	.2751485	0.65	0.520	-.3764041	.7333758	47	INTERVENTION
QUADS DOM	-.1468699	.0553319	-2.65	0.011	-.2580074	-.0357325	54	AND CLINICAL
SGRQI	.0816139	.2234533	0.37	0.716	-.3672054	.5304331	54	GROUP

MWT	1.078732	1.302553	0.83	0.412	-1.544745	3.702209	50	ADJUSTED FOR
SNIP	.181307	.2570293	0.71	0.484	-.337399	.7000131	47	INTERVENTION,
QUADS DOM	-.1478509	.0529411	-2.79	0.007	-.25424	-.0414617	54	CLINICAL
SGRQI	.0764789	.2163264	0.35	0.725	-.3582453	.5112032	54	GROUP AND AGE

MWT=distance in the 6MWT, SNIP=Sniff nasal inspiratory pressure, QUADS DOM=Quadriceps strength in the dominant leg and SGRQ-I=total score in the St George's Respiratory Questionnaire for ILD

There is some evidence to suggest that, at baseline, the DLCO is negatively associated with QUADS DOM but, based on these data, none of the other important measurements seemed to have been affected.

3. However, when analysing the data longitudinally, after adjusting for time and its quadratic term, interactions between time and intervention and clinical groups and age, the effect on the QUADS DOM seem to weaken. Below, we additionally adjust the models presented in Table 1 by DLCO – after investigating its interactions with intervention/clinical groups too. *Only coefficients corresponding to DLCO are shown for a succinct message, but they are adjusted for time, intervention groups and clinical groups, age and their interactions..*

Table 2: Relationship between DLCO and key outcomes, adjusted for time, intervention groups and clinical groups, age and their interactions.

	Estimate	SE	T value	p-value	95%CI low	95%CI high	No Obs	Association
MWT	1.336415	.9914012	1.35	0.178	-.6066961	3.279525	113/53	ADJUSTED FOR
SNIP	.3109333	.2361878	1.32	0.188	-.1519863	.7738529	113/53	INTERVENTION,
QUADS DOM	-.0910743	.0473067	-1.93	0.054	-.1837937	.0016452	119/55	CLINICAL
SGRQI	.1404341	.197889	0.71	0.478	-.2474212	.5282893	120/55	GROUP AND AGE

MWT=distance in the 6MWT, SNIP=Sniff nasal inspiratory pressure, QUADS DOM=Quadriceps strength in the dominant leg and SGRQ-I=total score in the St George's Respiratory Questionnaire for ILD

This is not to say that the effect on these outcomes may not be important, we just did not capture it based on these data.

3. **We know how many patients dropped out at each time point and did not have assessments, but what is known about how they engaged with the intervention? Also, significantly more participants dropped out of the high intensity intervention, so it is unclear whether the conclusion that it is feasible is substantiated.**

a) Engagement:

All participants included in this study were volunteers and engaged fully with the exercise-based pulmonary rehabilitation classes and the 6-month programme. They were closely monitored by the physiotherapists during the classes and had regular phone calls and a visit in the 6-month post-PR period. This is shown in the following paragraphs:

Page 6, lines 37-38 – *'This was an 8-week, twice weekly outpatient programme, including only people with ILD, which consisted of circuit-based training, supervised by two experienced physiotherapists.'*

Page 7, lines 43-58- *'Following completion of the PR programme, participants commenced a personalised community exercise programme, starting at the level they achieved at the end of PR. The 6-month programme included outdoor walking in HIIT or MICT mode, as well as home-based stretching and strength training exercises. Progress was monitored by phone calls (once per month) and at least one home visit by the research physiotherapists on the 3rd month. The physiotherapists guided participants on how to fill in the exercise diaries and progress training depending on mode of exercise. The ILD-PR booklet was used as a resource for the home exercises and as a reminder of the educational topics.'*

Overall, we found engagement to be very good but, as expected in all clinical trials, there are always variations in engagement when participants are not closely supervised, such as in the community-based training programme.

During that period, phone calls and home visits by the physiotherapists encouraged participants to adhere to their personalised programme and progress their training. As expected with all exercise programmes, benefits started to decline at some point.

Our pilot study offers valuable data about the point where benefits from PR start to decline in the 6-month period. This can help the design of future research so that more exercise, or supervision is offered at that point. We have highlighted this important issue under 'Implications for future studies':

Page 19, line 47-55: *'Our results also provide valuable information on the time point where 6MWD and other variables start to deteriorate post-PR and therefore, an estimate of time when additional PR support may be needed, thus optimising PR provision for ILD patients.'*

b) Feasibility:

Our conclusion regarding feasibility, refers to the feasibility of including HIIT exercises within a pulmonary rehabilitation programme specifically designed for patients with ILD (ILD-PR programme). This was described in our initial aims:

Page 5, lines 1-7: *'The aim of this pilot study was therefore to evaluate: a) the feasibility of using HIIT in an ILD-PR programme, b) determine the short and medium-term effects of HIIT on exercise capacity, respiratory and peripheral muscle strength, breathlessness and health status and c) explore responses in different types of ILD.'*

This was an important point for our pilot study, as it gave valuable insight in terms of resource allocation and acceptability of this mode of exercise by ILD patients, which will be useful for the design of a definitive study on HIIT in ILD rehabilitation. The HIIT group had a similar number of eligible patients who consented to the study, reasons for discontinuing the programme and no adverse events from exercise.

Our conclusion was that it is possible to include HIIT as part of PR, it is acceptable by patients with ILD and it is indeed possible to continue using this mode of exercise in a community-based programme. The remaining observations (regarding resource allocation and attrition in each group) were also reported in our manuscript so that we (and other researchers) can make informed decisions about the design of a future definitive study.

4. More details of the intervention are needed particularly the duration of intervals.

Thank you for your comment. We reported the average time on HIIT in the first week and explained how we progressed training.

Page 6, lines 47-54: *'In the HIIT group, intensity was set at 80% HRmax and patients were instructed to exercise in dynamic intervals followed by low intensity exercise or rest. The aim in this group was to increase the amount of time on the 'high intensity' phase each week. The average time on HIIT was 2.5 min in the first week and recovery time varied depending on type and severity of ILD. Progression of training was determined by reducing time in the low intensity phase of HIIT and/or increasing time in the target HR.'*

However, each patient had an individualised programme to ensure that oxygen saturation was kept above 85% while on HIIT. The duration of intervals, therefore, varied per person depending on oxygen saturation responses to exercise. Nevertheless, we can only understand what happens on average as it is usually subject to statistical analysis and inference. We have included more information about the total time on HIIT during the aerobic exercises that were included in our programme as shown here:

Page 7: *'In the first week, the average time on HIIT (at 80% HRmax) for each aerobic exercise was 2.5 min, therefore HIIT was 12.5 minutes of the total 30-minute session, and this time increased each week. Progression of training was determined by reducing time in the low intensity phase of HIIT and/or increasing time in the target HR . Each participant's individualised programme was recorded on exercise logs during the sessions.'*

5. The introduction and discussion should provide succinct context of the existing literature of the efficacy of moderate intensity training in PR for ILD and what gap/unanswered question this study examines.

Thank you for your suggestion. The role of Pulmonary Rehabilitation (PR) is well established across many respiratory conditions, including ILD. Exercise training during PR is usually delivered by moderate intensity continuous training, therefore the efficacy of this method is established by the results of studies in PR.

However, the number of studies on PR in ILD is limited and there is no data available about the optimal exercise mode. It is because of this lack of information on the optimal mode of exercise for ILD that we conducted the study.

We have included this information in the background and discussion and have added more information about the definition of MICT as well as HIIT.

6. Abstract- Participant sex and age by group are missing.

Thank you, this was indeed an omission on our part. We have now included the sex (M: F) and mean age (SD) of participants per group in the abstract.

7. Methods -Was a test 6MWD performed, as recommended by ATS guidelines?

Yes, the 6MWD test was performed as per ATS guidelines. We have included this statement and the reference on page 8.

8. Methods- Were PFTS (i.e. lung volumes) or CT scans used to assess the degree of restriction and extent of disease?

Yes, all participants were patients in the outpatient chest clinic of a tertiary referral hospital for Interstitial Lung disease. Prior to being included in this study, all participants had full lung function tests which were performed by specialised technicians in a lung function laboratory and CT scans were part of their clinical diagnostic process. This was part of their clinical management. Participants were also monitored throughout the study by respiratory physicians. The usual medical care was not affected by participation in this study.

We have now included a further clarification under 'Study population':

*'Symptomatic patients over 18 years old, of all types and severity apart from sarcoidosis and with a respiratory-physician diagnosis of ILD **from a tertiary referral hospital**, were invited to participate. This included patients with idiopathic pulmonary fibrosis (diagnostic criteria consistent with the International Consensus statement [1]), connective tissue disease related ILD, non-specific interstitial pneumonia, usual interstitial pneumonia, and chronic hypersensitivity pneumonitis.'*

9. Methods-Provide details of the education session topics.

Thank you for your suggestion. We have now included the topics discussed during the education sessions:

Page 7: *'The education session, on the 2nd hour of the programme was common for all participants and included presentations and discussion on topics important to ILD patients, **such as: Topics about breathing and breathlessness (mechanics of breathing, helpful/unhelpful breathing patterns, the Active Cycle of Breathing technique, useful positions for recovering, pacing), topics about ILD (what happens with ILD, cough, associated conditions-rheumatoid Arthritis, Gastro-Oesophageal Reflux disease, Sjögren's syndrome, Bird Fancier's Lung, Idiopathic Pulmonary Fibrosis) and topics about managing the condition (Why healthier lifestyle is important, keeping good posture, barriers to activity/exercise, dietary advice, smoking, incontinence, relaxation information, mood, anxiety and breathlessness, self-management and goal setting).'***

10. Results- Table 2 would be easier to understand if presented as a figure

Thank you for your comment. Table 2 is a very succinct presentation of all main measurements. It contains the predictions and their 95% CIs in terms of absolute values as well as in terms of changes both by intervention and clinical groups.

The changes are already presented as a figure – Figure 2 illustrates the changes from the baseline in the 4 main secondary outcomes by **intervention group** and Supplementary Figure 1 (in the **online supplement**) by **intervention and clinical group** (ILD subgroups as discussed on page 15, lines 34-47).

As this pilot study will be used to inform definitive studies, we believe it is important to keep both the table of actual values and the figures, so that other investigators can compare with their own results.

(Questions and suggestions from Reviewer 2)

11. In the eligibility criteria, the authors have mentioned "Patients were eligible to participate if they were ambulant and had symptoms of dyspnoea on exertion." What grade of dyspnoea was acceptable for inclusion in the study?

Thank you for your comment. We have included further details and reference about the grade of dyspnoea acceptable for inclusion.

Page 5, 'under Study population': *This included patients with idiopathic pulmonary fibrosis (diagnostic criteria consistent with the International Consensus statement [1]), connective tissue disease related ILD, non-specific interstitial pneumonia, usual interstitial pneumonia, and chronic hypersensitivity pneumonitis. Patients were eligible to participate if they were ambulant and had symptoms of dyspnoea on exertion (grades 1 to 3 in the mMRC scale (20)).*

12. Were participants having acute exacerbations also included? if yes, what was the criteria for inclusion.

Thank you for your question. Participants who were having acute exacerbations were not included in the study.

Our criteria for inclusion included ILD patients in stable condition.

13. The authors mentioned participants with LTOT in the results section. I think it would be good if it was mentioned initially in the eligibility criteria section for a clear perspective.

Thank you for this comment. We realise that we had not mentioned this point under eligibility criteria but included the number of participants in LTOT under Table 1 in our Results section. We have now included this under section 'Study population' - eligibility criteria:

Page 5- *'Patients were eligible to participate if they were ambulant and had symptoms of dyspnoea on exertion. Patients on long-term oxygen therapy (LTOT) were also included.'*

14. Did the authors decide regarding not including participants using supplementary oxygen during the PR program? I think inclusion of the same in the eligibility section would be necessary.

Thank you for your comment. Participants who were already on LTOT, continued to exercise with oxygen as recommended. However, we did not use additional oxygen for them during the exercise sessions and did not use supplementary oxygen for participants who were not already on LTOT.

15. The authors have mention "On the first day of the programme all participants were given an 'ILD-Pulmonary rehabilitation booklet' designed by our team, which contained information and pictures of all exercises, an exercise diary and summary of the education topics.". Was the educational booklet adapted from anywhere? if yes, it can be referenced accordingly. If not, how was the booklet developed, a short description regarding development and validation process should be mentioned in the methods section.

Thank you for your question and opportunity to elaborate on the development of the booklet.

All pulmonary rehabilitation programmes in the UK include an hour of education. This is a series of presentations and discussions on topics related to the disease(s) included in the programme. In our PR programme, we had a series of topics related to ILD. We have now included more information about them (please see answer in question 8).

Our 'booklet' was a summary of the key points in topics discussed. The booklet was put together by the two physiotherapists (included as authors in this manuscript) and the accuracy of the topics was checked by the health professionals who presented each session. For e.g. the topic on 'dietary advice for ILD' was checked by the dietitian in our hospital.

The booklet also included the exercises that were part of the PR programme, with pictures as reminders (the physiotherapists took pictures of themselves to show ILD patient the correct positioning during exercises). The end of the booklet included information about community gyms around South London and empty exercise logs so that patients in the home or community programme could follow the same programme as in the PR sessions.

The booklet was not an outcome measure, therefore did not need further 'validation' other than health professionals checking that the information was what they had discussed in the sessions. This booklet was a great resource for patients in the 6-month programme, as it reminded participants the order of exercises, correct posture and key points from the topics they had already heard in the education part of the PR programme.

The word limit of the manuscript does not allow further discussion of the booklet other than mentioning it as a source of information for patients.

Page 7, lines 56-58: '*The ILD-PR booklet was used as a **resource for the home exercises and as a reminder of the educational topics.***'

Presentation of the booklet does not add to the results of this pilot study but if you judge that it is required to be presented, we would be happy to add this information as an additional supplementary document.

16. How was the sample size determined? Detailed explanation regarding how the total number of ILD patients was derived needs to be mentioned.

Thank you for your question. This was a pilot study, therefore, not a hypothesis-testing study. Pilot studies do not require the same sample size calculation as definitive studies. An indicative sample size calculation was therefore based on the change in the distance in the 6 Minute Walk test (our primary outcome) as reported by previous studies.

Page 5, lines 25-33 'Statistical analysis':

'This was a pilot study to test feasibility and inform the design and power of a larger, definitive, randomised controlled trial. An intention to treat analysis was used. An indicative sample size calculation was based on previous studies aiming to detect between-group changes of 38± 43m with an 80% power at p <.05 significance level in the 6MWD following rehabilitation (26-28).'

17. How was the missing data managed? Were the drop-out rates considered?

Thank you for the question.

The missing data have been evaluated and, in inferential (hypothesis testing) settings, sensitivity analysis to the 'missing at random' assumptions would be conducted – the only solution as the assumption is not testable from the data at hand (see the above explanations).

Since this is a **pilot/feasibility study**, the secondary outcomes have been estimated under 'missing at random' assumption.

The following tables show the process that our statistician followed to consider the patterns of missing data overall (Table 1), patterns of missing data by intervention group (Table 2- Intervention group is the HIIT group and control group is the MICT group) and patterns of missing data by combined intervention and clinical group (Table 3). The statistician's conclusion about patterns of missing data is presented below Table 3 (in blue font).

The multivariate multiple imputation under the 'Missing at random assumption' is presented in Table 4 and the conclusion just below it.

There were 3 assessment points in this study: Baseline, post-PR and 6-months after the end of PR (or 8 months overall). The **+** symbol in the tables shows that data at this assessment exists, while the **dot symbol** '.' represents missing data in the particular assessment point.

Table 1: PATTERNS IN THE MISSING DATA OVERALL

PATTERN	MWT	SGRQI	SNIP	QUADS_DOM
+++	26	33	27	32
++.	6	2	3	2
+.+	1	1	1	2
..+	4	0	3	0
+..	19	21	18	20
..+	0	0	1	0
..+	0	1	1	1
...	2	0	4	1
TOTAL MISSING	32	25	31	26

Table 2: PATTERNS IN THE MISSING DATA BY INTERVENTION GROUP

PATTERN	CONTROL				INTERVENTION			
	MWT	SGRQI	SNIP	Q_DOM	MWT	SGRQI	SNIP	Q_DOM
+++	15	16	14	15	11	17	13	17
++.	1	1	1	1	5	1	2	1
+.+	1	1	1	2	0	0	0	0
.++	1	0	1	0	3	0	2	0
+..	6	7	6	7	13	14	12	13
..+	0	0	1	0	0	0	0	0
+..	0	0	0	0	0	1	1	1
...	1	0	1	0	1	0	3	1
TOT MISS	10	9	11	10	22	16	20	16

Table 3: PATTERNS IN THE MISSING DATA BY INTERVENTION AND CLINICAL GROUP

PATTERN	CONTROL								INTERVENTION							
	FIBROSIS				AUTOIMMUNE				FIBROSIS				AUTOIMMUNE			
	MW	SGR	SNI	Q_DO	MW	SGR	SNI	Q_DO	MW	SGR	SNI	Q_DO	MW	SGR	SNI	Q_DO
N	T	QI	P	M	T	QI	P	M	T	QI	P	M	T	QI	P	M
+++	3	4	3	4	12	12	11	11	8	11	8	11	3	6	5	6
++.	1	1	1	1	0	0	0	0	1	0	1	0	4	1	1	1
+.+	1	1	0	1	0	0	1	1	0	0	0	0	0	0	0	0
.++	1	0	1	0	0	0	0	0	2	0	1	0	1	0	1	0
+..	5	5	5	5	1	2	1	2	9	9	8	8	4	5	4	5
..+	0	0	1	0	0	0	0	0	0	0	0	0	0	0	0	0
+..	0	0	0	0	0	0	0	0	0	0	0	0	0	1	1	1
...	0	0	0	0	1	0	1	0	0	0	1	1	1	0	1	0
TOTAL MISS	8	7	8	7	2	2	3	3	12	9	11	9	10	7	8	7

The most common pattern of missing data (highlighted at the tables) is no different across intervention and clinical groups even by a simple chi-squared test for independent proportions ($p > 0.1$)

Table 4: Multivariate multiple imputation. The results do not differ from the complete data analysis, as expected as the mixed models inference operate under MAR (missing at random assumption).

MWT	Estimate	St Err	z-value	P-VALUE	95% CI - LOW	95% CI - HIGH
TIME	32.36228	6.712123	4.82	0.000	19.12189	45.60266
INTERVENTION GROUP	-48.35758	34.46726	-1.40	0.161	-115.9187	19.20358
TIME x INTERVENTION GROUP	-5.985328	3.10423	-1.93	0.055	-12.09925	.1285965
ILD GROUP	-27.47115	39.32687	-0.70	0.485	-104.5591	49.6168
TIME x ILD GROUP	1.287501	3.168639	0.41	0.685	-4.94634	7.521343
ILD GROUP x INTERVENTION GROUP	16.69733	53.18823	0.31	0.754	-87.59293	120.9876
TIME x ILD GROUP x INTERVENTION GROUP	6.909131	4.540814	1.52	0.130	-2.038378	15.85664
TIME SQ	-3.414554	.7027924	-4.86	0.000	-4.799801	-2.029307
AGE	-9.51875	2.177144	-4.37	0.000	-13.78686	-5.250645
AGE x INTERVENTION GROUP	3.67944	3.16184	1.16	0.245	-2.518378	9.877258
AGE x ILD GROUP	13.65491	3.875273	3.52	0.000	6.055673	21.25416

AGEx INTERVENTION GROUPx ILD GROUP	-11.52862	4.836897	-2.38	0.017	-21.01184	-2.045406
CONST	415.5424	27.94703	14.87	0.000	360.7613	470.3234
SGRQTI						
TIME	-4.843581	1.842466	-2.63	0.009	-8.472241	-1.214921
INTERVENTION GROUP	12.37265	7.316129	1.69	0.091	-1.971507	26.71681
TIME x INTERVENTION GROUP	.7682462	1.076266	0.71	0.477	-1.359176	2.895669
ILD GROUP	-3.328112	8.153141	-0.41	0.683	-19.31092	12.6547
TIME x ILD GROUP	.0597352	1.056929	0.06	0.955	-2.025384	2.144855
ILD GROUP x INTERVENTION GROUP	-12.45552	10.60683	-1.17	0.240	-33.24779	8.336751
TIME xILD GROUP x INTERVENTION GROUP	-.5952558	1.490742	-0.40	0.690	-3.539802	2.349291
TIME SQ	.5490916	.1954566	2.81	0.005	.1643347	.9338485
AGE	-.2979609	.4945523	-0.60	0.547	-1.269288	.673366
AGEx INTERVENTION GROUP	-.3754479	.6631105	-0.57	0.571	-1.675849	.924953
AGEx ILD GROUP	-.8462754	.8351287	-1.01	0.311	-2.48577	.7932188
AGEx INTERVENTION GROUPx ILD GROUP	1.273301	1.016754	1.25	0.211	-.721385	3.267987
CONST	42.30769	5.912391	7.16	0.000	30.71646	53.89891
SNIP						
TIME	3.543335	2.936514	1.21	0.229	-2.255126	9.341796
INTERVENTION GROUP	-1.926749	9.790125	-0.20	0.844	-21.13954	17.28604
TIME x INTERVENTION GROUP	-.3432158	1.927724	-0.18	0.859	-4.17076	3.484328
ILD GROUP	-9.211903	10.88005	-0.85	0.397	-30.55742	12.13362
TIME x ILD GROUP	1.768346	1.786657	0.99	0.324	-1.769603	5.306295
ILD GROUP x INTERVENTION GROUP	-12.67477	14.04343	-0.90	0.367	-40.22164	14.87209
TIME xILD GROUP x INTERVENTION GROUP	-.2600355	2.238869	-0.12	0.908	-4.687727	4.167656
TIME SQ	-.5423807	.329105	-1.65	0.101	-1.192705	.1079436
AGE	-1.230814	.6375214	-1.93	0.054	-2.484236	.0226073
AGEx INTERVENTION GROUP	.2631037	.9042342	0.29	0.771	-1.513268	2.039476
AGEx ILD GROUP	.6810118	1.120584	0.61	0.544	-1.522875	2.884899
AGEx INTERVENTION GROUPx ILD GROUP	-.8057667	1.373501	-0.59	0.558	-3.505065	1.893532
CONST	100.752	7.837596	12.85	0.000	85.37559	116.1283
QUADS_DOM						
TIME	1.769291	.6050814	2.92	0.004	.5780574	2.960524
INTERVENTION GROUP	-.7634722	2.118112	-0.36	0.719	-4.922818	3.395873
TIME x INTERVENTION GROUP	-.2609028	.3773727	-0.69	0.491	-1.007607	.485801
ILD GROUP	1.279066	2.243326	0.57	0.569	-3.121479	5.679612
TIME x ILD GROUP	-.2669902	.3926891	-0.68	0.498	-1.043498	.5095175
ILD GROUP x INTERVENTION GROUP	-1.379032	2.930769	-0.47	0.638	-7.127986	4.369922
TIME xILD GROUP x INTERVENTION GROUP	.2860477	.4773037	0.60	0.550	-.655852	1.227947
TIME SQ	-.1430013	.063486	-2.25	0.025	-.2678971	-.0181055
AGE	-.4186496	.1352376	-3.10	0.002	-.6847584	-.1525408
AGEx INTERVENTION GROUP	.2010724	.18352	1.10	0.274	-1.594132	.561558
AGEx ILD GROUP	.3787371	.2370755	1.60	0.111	-.0878907	.8453648
AGEx INTERVENTION GROUPx ILD GROUP	-.091465	.2820867	-0.32	0.746	-6.459109	.462981
CONST	18.36581	1.672908	10.98	0.000	15.08281	21.64881

*There are no strong violations for normality of the measurements. Shapiro Wilk tests for normality reveal $p=0.0225$ for *quads_dom0* and $p=0.01724$ for *snip0* but these values are not interpreted as evidence of strong departure from normality. **The imputation models are based on multivariate normal distribution assumption and do not reveal strong differences in the magnitude of the estimates or their precisions.***

18. The authors mention the study period for as "an 8-week, twice weekly outpatient programme" followed by a 6-month home exercise program. What was the schedule/ plan for the home exercise program? Did the patients perform the exercises all by themselves referring to the educational booklet or were monitored? If yes, how?

Thank you for your question. At the end of the PR programme, all participants were given a personalised exercise programme and were asked to continue to exercise 3 times per week as well as do daily walking. The plan was recorded in their own booklet. Some patients joined their local gym or organised exercise classes to ensure that they took part in group exercise at least once per week and followed a home programme the remaining time. Others opted for home-based exercise only. Regardless of the choice, during the 6-month period, patients were self-monitoring and had regular phone-calls and a visit from the physiotherapists. The aim during this period was to investigate how the groups would continue when self-managing in their own environment and observe the impact on outcome measures (presented in our figures).

(Questions and suggestions from Reviewer 3)

19. Please add the information on how to do HIIT and MICT in the interventions part of abstract.

Thank you for your suggestion. The strict word-count of the abstract does not allow further elaboration on how to do HIIT and MICT. However, we have included further information about MICT (as well as HIIT) in the 'background' and in the 'Intervention' part of our main text. Please see updated 'Background' and our answer to question 3.

20. The authors mentioned "Pulmonary rehabilitation (PR) is a well-established rehabilitation option for various respiratory conditions [5, 6] but in ILD patients the optimal training programme, especially for longer term benefits, remains unclear. ". are there other therapies for ILD? What are the effects?

Thank you for your question. There is no known 'therapy' for ILD as this is a progressive disease. The only possible option as 'therapy' is lung transplantation. There only 2 anti-fibrotic medications for this disease (Pirfenidone and Nintedanib) (see: Vincent Cottin, Lutz Wollin, Aryeh Fischer, Manuel Quaresma, Susanne Stowasser, Sergio Harari European Respiratory Review 2019 28: 180100; DOI: 10.1183/16000617.0100-2018)

As discussed under our eligibility criteria, our participants were at optimal medical management and under one of the two available medications. These medications did not interfere with the ability of patients to exercise.

21. Lack of the information on why the authors did this study.

Thank you for your question. This was a pilot, feasibility study and we conducted this study to generate data that would help us, and other researchers, to conduct larger, definitive studies on HIIT programmes for ILD patients. We have included this information under the following:

- Under **Abstract-Conclusion**- we had the following sentence: *'A definitive, multi-centre randomised controlled trial is required to definitively address the role of HIIT in ILD.'*
- Under **'Strengths and Limitations of this study'** we had the following clarification:
'Strengths and Limitations of this study'
 - *This randomised controlled feasibility, pilot study, showed that HIIT is feasible in circuit-based PR programmes.*
 - *This study adds valuable information for the design of future, definitive studies, on the peak and subsequent deterioration of outcomes over time depending on exercise mode.*
 - *Our results identify the point at which exercise reinforcement may be needed after completion of a HIIT programme.'*

We had also included a section called *'Implications for future definitive studies'* (page 19) under *'Discussion'* which highlights that we conducted this pilot study as a precursor to a definitive study.

22. The authors just introduce HIIT in the background section, but what is MICT?

Thank you for noticing that we hadn't included a definition of MICT. We have now included more information in the background to clarify this:

Page 4: *'The traditional exercise mode in PR is moderate intensity continuous training (MICT) which involves continuous exercise for 30-60 minutes at intensities ranging from 60-80% maximum heart rate. However, other training modalities may offer greater benefits.'*

23. The aim of this pilot study seems not to match the title.

Thank you for your comment. The aims of the study were expanded to include all individual elements we were investigating in this pilot, feasibility study. The title has a word limit and needed to be more concise.

We therefore opted for '**Comparison of...**' as an inclusive title for the different elements we investigated.

24. How did the authors do the randomization?

Thank you for your question. We performed stratified randomisation, according to ILD type to ensure balance of types between the 2 groups. Randomisation was led by the research physiotherapists and the researcher performing the assessments was blinded to the group participants belonged to.

Page 6, lines 11-26 under 'Randomisation and blinding':

'Following baseline assessments, participants were randomised into 2 groups: exercise using moderate intensity continuous training (MICT group) or exercise using high-intensity interval training (HIIT group). Participants were stratified by ILD type to ensure a balanced distribution of types between groups. They were placed into 3 sub-groups: a) Idiopathic group (e.g. IPF); b) Autoimmune group (CTD-ILD) and c) Chronic Hypersensitivity pneumonitis group (CHP/EAA). The investigator performing all assessments and the statistician were blinded throughout the study.'

25. Please give a reference for MICT was 60% maximum heart rate and HIIT was 80%. Or please give a clear definition for MICT and HIIT.

Thank you for your comment. We have included further definitions in the main text. Please also see our answer of question 18.

26. The authors mentioned "The average time on HIIT was 2.5 min in the first week and recovery time varied depending on type and severity of ILD", so how long did that and how to recommend for more patients in the clinic?

Thank you for your comment. We reported the average time on HIIT at the starting point (first week) and the total time on aerobic training. Progression of exercise per week meant that time on HIIT increased. We have included more information to elaborate this point.

Page 7: *'In the first week, the average time on HIIT (at 80% HRmax) for each aerobic exercise was 2.5 min, therefore HIIT was 12.5 minutes of the total 30-minute session, and this time increased each week. Progression of training was determined by reducing time in the low intensity phase of HIIT and/or increasing time in the target HR.'*

In relation to clinical recommendations, pilot studies are not hypothesis-driven definitive studies and do not result in clinical recommendations or final conclusions. The aim of pilot studies is to collect data that would inform the design and set up of a definitive study. Therefore, there can be no recommendations at this point in relation to the recommendations clinicians would make to patients in the clinic.

27. Who and where did the patients monitor the pulse oximeters and perceived exertion and breathlessness?

Thank you for your question. During the pulmonary rehabilitation programme, all patients were given pulse oximeters to wear and were monitored by the research physiotherapists (co-authors in this manuscript). The perceived exertion and perceived breathlessness were monitored using the BORG Rate of Perceived Exertion scale and the BORG Modified Breathlessness Scale respectively. These are well validated scales and are used in all pulmonary rehabilitation programmes in the UK and other countries. The information was recorded in the exercise logs provided during the session.

Page 7: *'Heart rate and oxygen saturation were monitored using pulse oximeters and perceived exertion and breathlessness were recorded at set intervals using the modified 10-point Borg scale and the Rate of Perceived Exertion scale (RPE) respectively [21, 22].'*

In the 6-month programme, participants had access to pulse oximeters and were self-monitoring and self-recording their data, including perceived exertion and breathlessness, in the exercise logs at the end of their booklet.

28. How to calculate the sample size?

Thank you for your question. Please see our answer to question 15.

29. Lack of the information on severity of ILD of the included patients

Thank you for your question. The severity of ILD is reported in our *'Table 1: Baseline characteristics'* (Page 11) where we presented each group's ILD-GAP index (**higher number signifies greater severity**) and Lung function (FVC (% predicted) and DLCO (% predicted)). These parameters reflect severity of ILD.

30. The authors mentioned all types and severity of ILD would included in this study, so I just want to know how the authors did the risk monitoring and what they did for the potential danger?

Thank you for your question and the opportunity to elaborate. All participants were recruited via the chest clinic and had optimal medical management before commencing the exercise programme. Our eligibility criteria ensured that patients who may have been at risk from exercise were excluded.

During the exercise programme, risk was monitored by the physiotherapists delivering the programme. All qualified and registered physiotherapists in the UK are equipped with knowledge and skills to guide people with respiratory diseases and common comorbidities such as diabetes and hypertension, through exercise programmes. The exercise programme took place in a gym within a hospital. In the hospital setting, resuscitation equipment and oxygen are readily available, and the physiotherapists were trained to use the equipment and respond to emergencies (basic life support training). The physiotherapists also guided participants on how to monitor themselves before, during and after each exercise session.

Oxygen saturation levels were monitored throughout training. Stretching exercises and warm up and cool-down were part of the programme and minimised risk from musculoskeletal injuries or cardiac-related adverse events.

In the community setting, where patients were exercising at home or in local centres, a more pragmatic approach is taken. Participants were trained to monitor desaturation level, stretching exercises were given to avoid minor musculoskeletal injuries and there was an opportunity for participants to call the physiotherapists if they needed to discuss concerns about the exercises. As discussed in question 26, participants were trained to monitor their own perceived exertion and breathlessness, therefore, minimised risk during their community-based programme.

Our study monitored closely any adverse events and recorded them. No adverse events were reported from exercise in either group.

31. How about to take medicines for the comorbidities?

Thank you for the question. Clinical studies include participants who take medicines for comorbidities. This is inevitable as clinical studies are pragmatic studies, especially they include older participants, as is the case in people with ILD.

The medications taken did not interfere with exercise. The only exception was prednisolone, that is a steroid and may have an effect on muscle function. This is the reason we reported the use of prednisolone per group in Table 1 and have reported the level (in mg).

Page 11, lines 18-27 :

Prednisolone medication (no. (%))				
• < 5mg	19 (32.8)			0.34 ^s
• 5-10 mg	19 (32.8)	9 (36)	10 (30.3)	
• >10 mg	3 (5.17)	5 (20)	14 (42.4)	
		2 (8)	1 (3)	

32. Did the included patients take medicines for ILD?

Thank you for your question. Please see our answer in question 19.

33. The reference would be up-to-date.

Thank you for your comment. The authors presume you refer to the references that we added in the reference list. We have checked and updated references on guidelines and Cochrane reviews to their more recent version.

VERSION 2 – REVIEW

REVIEWER	Wang, Zhijie Guangzhou University of Chinese Medicine, Medical College of Acupuncture and Rehabilitation
REVIEW RETURNED	21-Apr-2023
GENERAL COMMENTS	Thank you for inviting.